# 3D Melanoma Cocultures as Improved Models for Nanoparticle-Mediated Delivery of RNA to Tumors

**DOI:** 10.3390/cells11061026

**Published:** 2022-03-17

**Authors:** Maximilian E. A. Schäfer, Florian Keller, Jens Schumacher, Heinrich Haas, Fulvia Vascotto, Ugur Sahin, Mathias Hafner, Rüdiger Rudolf

**Affiliations:** 1Institute of Molecular and Cell Biology, Hochschule Mannheim, 68163 Mannheim, Germany; m.schaefer@doktoranden.hs-mannheim.de (M.E.A.S.); f.keller@hs-mannheim.de (F.K.); m.hafner@hs-mannheim.de (M.H.); 2Biopharmaceutical New Technology (BioNTech) SE, 55131 Mainz, Germany; jens.schumacher@biontech.de (J.S.); heinrich.haas@biontech.de (H.H.); ugur.sahin@biontech.de (U.S.); 3Translational Oncology (TRON), University Medical Center, Johannes Gutenberg University Mainz, 55131 Mainz, Germany; fulvia.vascotto@tron-mainz.de; 4Institute of Medical Technology, Heidelberg University and Hochschule Mannheim, 68163 Mannheim, Germany; 5Center for Mass Spectrometry and Optical Spectroscopy (CeMOS), Hochschule Mannheim, 68163 Mannheim, Germany

**Keywords:** lipoplex, mRNA, nanoparticles, cancer, tumor targeting, tumor models, in vitro in vivo correlation (IVIVC)

## Abstract

Cancer therapy is an emergent application for mRNA therapeutics. While in tumor immunotherapy, mRNA encoding for tumor-associated antigens is delivered to antigen-presenting cells in spleen and lymph nodes, other therapeutic options benefit from immediate delivery of mRNA nanomedicines directly to the tumor. However, tumor targeting of mRNA therapeutics is still a challenge, since, in addition to delivery of the cargo to the tumor, specifics of the targeted cell type as well as its interplay with the tumor microenvironment are crucial for successful intervention. This study investigated lipoplex nanoparticle-mediated mRNA delivery to spheroid cell culture models of melanoma. Insights into cell-type specific targeting, non-cell-autonomous effects, and penetration capacity in tumor and stroma cells of the mRNA lipoplex nanoparticles were obtained. It was shown that both coculture of different cell types as well as three-dimensional cell growth characteristics can modulate distribution and transfection efficiency of mRNA lipoplex formulations. The results demonstrate that three-dimensional coculture spheroids can provide a valuable surplus of information in comparison to adherent cells. Thus, they may represent in vitro models with enhanced predictivity for the in vivo activity of cancer nanotherapeutics.

## 1. Introduction

Delivery of mRNA by lipid-based nanoparticles enabled the development of a new generation of therapeutic agents [1], which allowed not only the development of prophylactic vaccines for SARS-CoV-2 coronavirus, but is also in development for various other types of therapeutic intervention, including the therapy of different cancer entities [2,3,4,5]. For most of these applications, delivery of the nanoparticles to certain organs or cellular compartments is required. For example, it has been shown that lipoplex (LPX) nanoparticles can be engineered for targeting either lung or spleen, depending on their physicochemical characteristics such as size and charge [5,6]. In the spleen, the main target has been antigen-presenting cells (APCs), which enabled development of a technology platform for application in cancer immunotherapy, with several products currently being evaluated at different stages of clinical studies [5,6,7]. The LPX are used for treating different cancer indications, including melanoma. Several other applications of the LPX are currently being tested in clinical studies or are in preparation (undisclosed).

Other approaches of therapeutic intervention in cancer require direct targeting of the cargo to the tumor. While such tumor targeting is already a challenge for delivery of small molecules, with mRNA as therapeutically active compound the complexity is even higher, as not only accurate control of physical targeting, but also successful uptake and translation into the encoded protein by the targeted cells must be accomplished. Depending on the intended therapeutic intervention, different cell types in the tumor microenvironment need to be targeted. In such cases, an accurate control of the target cells and a better understanding of uptake mechanisms are required for designing the RNA therapeutics.

Typically, such information can be obtained only with in vivo studies, as the usual cell culture models are not sufficiently predictive for the complex 3D cellularity in the tumor. However, the extensive screening needed to elucidate coherence between formulation characteristics and targeting at the cellular level exceed the possibilities of in vivo studies. Therefore, improved in vitro models, which are predictive for the situation in vivo, would be extremely helpful to facilitate development and screening of such pharmaceutical products.

We here addressed cell-type specific targeting, non-cell-autonomous effects and penetration capacity of mRNA lipoplex nanoparticles using the translation of the reporter protein, eGFP, after application of the LPX to 2D and 3D cell models of melanoma.

Cutaneous melanoma is a highly aggressive form of skin cancer deriving from the transformation of pigment-producing melanocytes, which are normally found in the basal layer of the epidermis [8]. With increasing progression of the disease, radial growth phase, vertical growth phase, and metastatic melanoma can be distinguished [9,10,11]. These three phases are characterized by lateral spread, occasional passage through the basement membrane, and dissemination through the bloodstream, respectively. Surgical removal of the tumor is only feasible until the radial growth phase and the early vertical growth phase [12]. Treatment of subsequent stages, in general, is known for low response rate and multidrug resistance [13].

The applied 3D cell model, which is composed of CCD-1137Sk fibroblasts, HaCaT keratinocytes and SK-MEL 28 melanoma cells [14,15], was previously shown to recapitulate basic features of early melanoma, i.e., radial growth phase and early vertical growth phase, including loss of keratinocyte differentiation, melanoma cell invasion, and cytostatic-induced increase of ABCB5 expression in external melanoma cells [14]. In the present study, treatment with mRNA lipoplex nanoparticles revealed distinct effects on eGFP translation levels of the different cell types, depending on the spatial 3D structure and the complexity of the systems.

## 2. Materials and Methods

### 2.1. Assembly of LPX

LPX were assembled using protocols as described earlier [5,6]. Briefly, mRNA encoding for eGFP, provided in a HEPES/EDTA buffer (10 mM/0.1 mM) was conditioned with a sodium chloride solution to arrive at a sodium chloride concentration of 224 mM and a RNA concentration of 0.2 mg/mL (~0.66 mM). The RNA (unmodified) was manufactured at BioNTech (Mainz, Rhineland-Palatinate, Germany) using internal protocols. The conditioned RNA was mixed one-to-one (*v*/*v*) with cationic liposomes consisting of DOTMA (R-1,2-di-O-oleoyl-3-trimethylammonium propane) as a cationic lipid and DOPE (1,2-Dioleoyl-sn-glycero-3-phosphoethanolamine) as helper lipid in a 2:1 molar ratio. Liposomes were manufactured at BioNTech with a proprietary protocol derived from the ethanol injection technique. The concentration of the DOTMA in the liposomes was 0.284 mg/mL (~0.42 mM), corresponding to a molar ratio DOTMA/RNA (calculated as one negative charge per nucleotide, 33 Da) of about 0.65. The molar (charge) ratio of DOTMA to RNA inside the LPX was about one to one [16]. Because the LPX were assembled at an excess of RNA regarding charge ratio, an equivalent fraction of free, uncomplexed RNA (~35%) was present in the formulations. The LPX were compact globular particles and were characterized by a distinct internal lamellar organization consisting of repeating lipid bilayers where the RNA was inserted into the hydrophilic slab in between the adjacent bilayers [5,7,16]. For quality control, the sizes of the liposomes and the LPX were determined by using dynamic light scattering measurements (Nicomp ZLS Z3000, Santa Barbara, CA, USA). Concentration of the lipids in the liposome was controlled by RP-HPLC (Agilent). Size measurements from two independent manufactured batches yielded results as follows:

Size (nm)Polydispersity indexLiposomes3500.35Lipoplex Batch 12640.14Lipoplex Batch 22660.12

RNA concentrations in the starting phase and the final lipoplex formulation were determined by UV-vis measurements (Nanodrop, Thermo Fisher Scientific Inc., Waltham, MA, USA), RNA integrity was controlled by capillary electrophoresis measurements (Fragment Analyzer, Agilent Technologies Deutschland GmbH, Waldbronn, Baden-Württemberg, Germany), typical integrity values were >95%. Prior to the measurements described here, the activity of the LPX was controlled by internal standard cell culture measurements at BioNTech.

### 2.2. Cell Culture and Lipoplex Nanoparticle Treatment

For 2D as well as spheroid cultures, human fibroblast CCD-1137Sk (ATCC^®^ CRL-2703™), human keratinocyte HaCaT (CLS order no. 300493) and human melanoma SK-MEL 28 (CLS order no. 300337) cell lines were used. SK-MEL 28 and HaCaT cells were cultured in Dulbecco’s Modified Eagle Medium (DMEM) with high glucose (4.5 g/L), L-Glutamine, and sodium pyruvate (Capricorn, Scientific GmbH, Ebsdorfergrund, Hessen, Germany), supplemented with 10% fetal bovine serum (Sigma-Aldrich, St. Louis, MO, USA) and 1% Penicillin Streptomycin (Capricorn, Scientific GmbH, Ebsdorfergrund, Hesse, Germany). CCD-1137Sk cells were cultured in Iscove’s Modified Dulbecco’s Medium (IMDM) with L-Glutamine, supplemented with 10% fetal bovine serum (Sigma-Aldrich, St. Louis, MO, USA), and 1% Penicillin Streptomycin (Capricorn, Scientific GmbH, Ebsdorfergrund, Hessen, Germany). All cells were maintained at 37 °C in 5% CO_2_. On day 0 of each experiment, cell viability was determined (Vi-CELL XR, trypan blue method, Beckman Coulter Inc., Brea, CA, USA). Routinely, cells were tested for mycoplasm using the MycoAlert Mycoplasm Detection Kit (Lonza Group AG, Basel, Basel city, Switzerland). All 2D and 3D experiments were performed in mono- and cocultures. Transfections with LPX nanoparticles and RNA encoding eGFP were executed in serum-free medium for a period of 24 h at 37 °C in 5% CO_2_. The LPX concentration range used for transfection of the 2D cultures was based on previous reports and adapted to the different cell lines used in this study [5,16]; (NCT02410733). As reference concentration for 2D cultures, 1.25 ng/µL of the LPX was used. The 3D cultures required 10 ng/µL LPX for transfection.

### 2.3. 2D/3D Mono- and Cocultures

The preparation of 2D cultures required 96-well flat-bottom microplates (Greiner AG, Kremsmünster, Upper Austria, Austria). For 2D mono- as well as for 2D cocultures, a total number of 3 × 10^4^ viable cells per well (viability > 96%) were seeded. In each experiment, the cell number was determined using the Vi-CELL XR (Beckman Coulter Inc., Brea, CA, USA), trypan blue method. Then, cells were cultured for 24 h before they were washed (3 × 5 min, PBS, Sigma-Aldrich, St. Louis, MO, USA) and treated with the LPX in serum-free medium. Spheroids were generated using 96-well cell repellent plates (Greiner AG, Kremsmünster, Upper Austria, Austria). For 3D monocultures, 1 × 10^4^ viable cells per well (viability > 96%) were seeded. After seeding, the well plates were centrifuged (5 min, 34× *g*) to allow for rapid spheroid formation. Then, spheroids were cultured for 48 h before treatment with LPX in serum-free medium for a period of 24 h at 37 °C in 5% CO_2_.

Triculture spheroids composed of CCD-1137Sk, HaCaT, and SK-MEL 28 cells were prepared in 96-well plates with a cell-repellent bottom (Greiner AG, Kremsmünster, Upper Austria, Austria) in two steps: First, 1 × 10^4^ fibroblasts per well (viability > 96%) were seeded, centrifuged (5 min, 34× *g*) and cultured for 72 h. In the second step, HaCaT (1 × 10^4^ cells/well) and SK-MEL 28 cells (2.5 × 10^3^ cells/well, viability > 96%) were added together and centrifuged as before. After coculturing for 48 h, the spheroids were transferred to serum-free medium and treated with LPX (PBS as negative control) for 24 h. The creation of the triculture was strictly based on [14]. After the 24 h LPX treatment, 3D monocultures and tricultures were washed with PBS (3 × 5 min) and fixed with paraformaldehyde solution (4% in PBS, RT, 30 min). For further processing, the spheroids were transferred to 1.5 mL tubes.

### 2.4. Live-Cell Imaging and Optical Clearing

Immediately after incubation with LPX, 2D mono- and cocultures were washed with PBS and nuclei were stained with DRAQ5 (Invitrogen, Carlsbad, CA, USA, diluted 1:1000) for 30 min at 37 °C. Analysis of GFP fluorescence in 2D cultures used live-cell confocal microscopy (see below). Analysis of eGFP translation in spheroids was performed in fixed and optically cleared whole mounts. After fixation with paraformaldehyde solution (Carl Roth, Karlsruhe, Baden-Württemberg, Germany) (4% in PBS, RT, 30 min) the spheroids were quenched with 0.5 M glycine (Carl Roth, Karlsruhe, Baden-Württemberg, Germany) in PBS for 1 h at 37 °C. Next, they were incubated in penetration buffer (0.2% Triton X-100, 0.3 M glycine, and 20% DMSO all Carl Roth, Karlsruhe, Baden-Württemberg, Germany) in PBS for 30 min at RT. Then, spheroids were washed with PBS/1% FBS and incubated in blocking buffer (0.2% Triton X-100, 1% bovine albumin serum (BSA, Carl Roth, Karlsruhe, Baden-Württemberg, Germany), 10% DMSO in PBS) for 2 h at 37 °C. For nuclear staining, DRAQ5 was incubated overnight at 37 °C with gentle shaking. Subsequently, samples were washed 5 times for 5 min in washing buffer at RT. Finally, samples were washed with ddH_2_O and refractive index was adjusted to 1.456 with Dimethyl sulfoxide (Carl Roth, Karlsruhe, Baden-Württemberg, Germany), urea (Nacalai Tesque Inc., Kyoto, Kyoto Prefecture, Japan), quadrol (Tokyo Chemical Industry, Tokyo, Tokyo Prefecture, Japan), sucrose (Nacalai Tesque Inc., Kyoto, Kyoto Prefecture, Japan) and glycerol (88%) (Carl Roth, Karlsruhe, Baden-Württemberg, Germany) for 48 h at constant RT followed by mounting on 18 well μ-slides (Ibidi GmbH, Gräfelfing, Bavaria, Germany) in the same solution. Procedures were adapted from recent protocols [17,18].

### 2.5. Data Acquisition and Analysis

Micrographs were taken with an inverted Leica TCS SP8 confocal microscope equipped with Leica Application Suite X software (both Leica Microsystems CMS GmbH, Mannheim, Baden-Württemberg, Germany) and 488 nm and 633 nm lasers for GFP and DRAQ5 excitation, respectively. For all images of 2D and spheroid cultures, 5×/0.15 and 10×/0.3 Fluotar objectives were used, respectively. Images of all samples within an experiment were acquired with the same setting. Quantitative image analysis was performed using ImageJ (version 1.52p, National Institutes of Health, Bethesda, MD, USA). Therefore, the areas of eGFP fluorescence signals were threshold-adjusted to the negative control. Data were normalized to the total area of DRAQ5 signals, i.e., the eGFP-positive area was determined as a ratio of nuclear fluorescence signal area. For penetration experiments, a single optical plane taken in the center of the spheroids was analyzed. For the analysis of the ring-like regions, the ImageJ (version 1.52p, National Institutes of Health, Bethesda, MD, USA) erosion function was used on single optical planes. This allowed four areas (outer ring, mid ring, inner ring and core) to be defined that represented the spheroids from the outer border to the core. Using a threshold, the eGFP signal area was determined for each part and normalized to the area of the nuclei signal. The amount of DRAQ5 staining per microscopic field showed that the number of cells in the various conditions did not vary significantly (Appendix A).

All graphs were created with GraphPad Prism (version 8.0.1, GraphPad Software, San Diego, CA, USA). The data are presented as mean ± SEM and statistically analyzed using two-way ANOVA with post hoc Tukey’s test (2D data), one-way ANOVA with post hoc Tukey’s test or Student’s *t*-test. *p*-values are indicated as * < 0.05, ** < 0.01, *** < 0.001, **** < 0.0001.

## 3. Results

### 3.1. Approach for Comparing the Different Cell Culture Models

To address coherencies and differences between 2D and 3D cell cultivation in single and combined culture, mono- and tricultures of fibroblasts, keratinocytes, and melanoma cells were investigated as 2D and 3D cell-culture models. For all systems, transfection efficacy was investigated using the same type of LPX formulation comprising mRNA encoding for eGFP in order to facilitate comparability (see Figure 1 for an outline of the testing procedure). Protocols, as previously used to transfect 2D monoculture cells, were adapted and applied [6,16]. Cell-type-specific differences in combination with cultivation type were revealed as outlined subsequently.

### 3.2. In 2D Culture, LPX-Mediated eGFP Expression Was Highest for Tricultures and HaCaT Cells

Previous studies provided initial information about the targeting characteristics of mRNA nanoparticles in different mouse models [6]. In the present work, the efficiency of transfection was first tested in 2D cultures and compared between mono- and tricultures of fibroblasts, keratinocytes, and melanoma cells. Qualitative visual inspection of the cell cultures did not reveal alterations in cell morphology or the presence of dead cells under the used LPX dosages (not shown). In addition, cell numbers did not vary significantly for the different conditions (Appendix A), indicating a good tolerability of the applied LPX. Microscopic analysis of GFP-fluorescence intensities and corresponding areas showed a differential reporter expression depending on cell type and LPX dosage (Figure 2A). Indeed, eGFP expression was highest in tricultures at the lowest tested LPX dose. Of the monocultures, HaCaT cells showed the strongest eGFP signals. On average, SK-MEL 28 and CCD-1137Sk displayed lower eGFP signals. This was confirmed by quantitative image analysis (Figure 2B). In summary, tricultures of fibroblasts, keratinocytes, and melanoma cells showed a trend towards the highest reporter expression under all tested LPX dosages, suggesting a slightly synergistic activity for LPX uptake and/or expression in the cocultures.

### 3.3. In Monoculture Spheroids, Melanoma Cells Showed the Highest eGFP Expression

To investigate the transfection efficacy in 3D cultures, eGFP expression was monitored first in monoculture spheroids of CCD-1137Sk, HaCaT and SK-MEL 28 cells. Therefore, cells of each cell line were cultured as spheroids, treated with LPX and then analyzed by 3D confocal microscopy (Figure 3). Similar to the situation in adherent 2D cell culture, spheroids derived from HaCaT cells showed a comparably high level of eGFP expression. In contrast to the 2D cultures, spheroids composed of CCD-1137Sk cells displayed the lowest eGFP signals and SK-MEL 28 spheroids were similarly well transfected as HaCaT spheroids. Furthermore, each cell line showed an individual in situ expression pattern. While eGFP expression was distributed all over the entire SK-MEL 28 spheroid, the denser CCD-1137Sk and HaCaT spheroids exhibited their eGFP signals rather in the periphery (Figure 3).

### 3.4. eGFP Signal Distribution Showed Cell-Type Specific Differences in Monoculture Spheroids

To analyze the differential spatial eGFP signal patterns in the monoculture spheroids, a segment analysis was performed. This determined the relative eGFP signal intensity from the spheroid border to its center. Single optical slices at the largest spheroid diameter were segmented into four ring-layers (one outer ring, one inner core, and two rings in between). Since spheroids of the different cell types varied in size, the wall thicknesses of the rings were adjusted to the average size of spheroids for a given cell type. Within each segment, the eGFP signal area normalized to nuclear signal area was determined for the respective cell types (Figure 4). This showed that in HaCaT and CCD-1137Sk spheroids, eGFP signals were largely limited to the outer edge of the spheroids (one or two exterior rings). In contrast, the SK-MEL 28 spheroids showed eGFP fluorescence in all layers with a tendency to higher expression towards the spheroid core. Altogether, these data suggested that efficiency and distribution of transfection were dependent on cell type and/or compactness of the monoculture spheroids.

### 3.5. Melanoma Triculture Spheroids Showed Reduced eGFP Expression

To assess the transfection behavior in a complex cellular environment, where different cell types are present in a nonuniform distribution, as likely found in situ, we examined triculture spheroids, composed of CCD-1137Sk cells, HaCaT cells and SK-MEL 28 cells, upon LPX exposure. From previous studies, the overall organization of the triculture spheroids was known to consist of a core formed by fibroblasts surrounded by keratinocytes. Furthermore, the melanoma cells were known to be localized in patches on the exterior of the spheroids, and this was confirmed by cell-type specific staining and fluorescence microscopy (Appendix A). SK-MEL 28 cells were also characterized by larger cell nuclei and a more sparse arrangement as compared to the denser cultures of fibroblasts and keratinocytes [14,15]. This observation was useful to assign the eGFP signals upon LPX treatment to the individual cell types and the total amount of eGFP expression in the triculture spheroids was quantified (Figure 5). Notably, a lower overall eGFP expression than for any of the monoculture spheroids was observed (compare Figure 3).

### 3.6. In Triculture Spheroids, eGFP Signals Were Largely Confined to the Outer Rings

To assess the eGFP signal distribution pattern in triculture spheroids, the segmental area analysis with ring and core regions of interest was performed (Figure 6A). This showed that most of the eGFP-positive cells were located in the outer and middle ring segments. However, other segments also contained some eGFP-positive cells. Quantitative analysis confirmed an overall low level of eGFP-positive regions (Figure 6B) as compared to the monoculture spheroids (compare to Figure 4D–F).

## 4. Discussion

There is a need for predictive in vitro model systems to allow systematic and comprehensive screening of suitable delivery vehicles [19]. So far, cell culture models lack predictivity, presumably because the standard model systems do not adequately reflect the complex situation in the tumor. This study addressed the uptake and reporter gene expression of RNA lipoplex nanoparticles in complex in vitro melanoma tumor models, namely containing tumor and stroma cells in 2D and 3D cell cultures. By using different combinations in mono- and cocultures, the lipofection behavior was found to be dependent on the cellular as well as the 3D context.

While in adherent (2D) monocultures, LPX-mediated eGFP expression was low and did not differ significantly between fibroblasts (CCD-1137Sk), keratinocytes (HaCaT), or melanoma (SK-MEL 28) cells, transfection efficiency was significantly increased in 2D tri-cultures. According to present concepts of Warburg [20] and Reverse Warburg effect [21,22], which can be observed in 70–80% of cancers [23], interactions between fibroblasts and cancer cells can have stimulatory effects on proliferation and metabolic interactions in different types of cancers [24,25], such as melanoma [26]. Therefore, one may hypothesize that the coculturing of fibroblasts, keratinocytes, and melanoma cells in this study affected their mutual metabolic behavior, including the intensity of cellular uptake.

Using 3D monoculture spheroids, results differed substantially from those obtained with the 2D models. While transfection efficiencies in adherent monocultures were not significantly different between the three tested cell lines, spheroids composed exclusively of SK-MEL 28 cells showed the highest transfection rate of all cell lines. Evidence for this came from both microscopic (Figure 3) as well as cytometric analysis (Appendix A). It cannot be excluded that this was, at least partially, due to an enhanced up-/down-regulation of uptake into SK-MEL 28 or CCD-1137Sk cells, respectively. However, penetration of the reagent into the spheroids was also likely affected; whereas SK-MEL 28 spheroids showed eGFP-positive cells throughout the spheroid width, eGFP signals were limited to the outer layers in both HaCaT and CCD-1137Sk spheroids. A likely explanation for this restriction of GFP fluorescence is that CCD-1137Sk and HaCaT cells formed rather compact and tight layers, while SK-MEL 28 cells generated more loose spheroids, which could be related to their decreased E-cadherin expression [27]. Morphologically, this was corroborated by small and densely clustered cells in spheroids composed of CCD-1137Sk and HaCaT cells and larger cells in the case of SK-MEL 28 spheroids (Figure 3). In this context, the differential cell sizes might hint to the discussed variations in metabolic and proliferative activity. Additionally, previous work showed that both CCD-1137Sk and HaCaT cells have a low proliferation capacity in spheroids and that proliferating cells were almost exclusively found in the outer regions of spheroids [14]. This was confirmed by additional immunofluorescence-staining data using the proliferation marker, Ki67 (Appendix A). In summary, location in a spheroid, cell density, and metabolic/proliferative activity might be important additional determinants of lipofection efficiency.

Finally, we observed that triculture spheroids composed of a CCD-1137Sk core, a HaCaT ring, and outer groups of SK-MEL 28 cells (originally described in [14,15], confirmed in Appendix A), showed an overall reduction in eGFP-positive cells compared to the 2D tricultures and also compared to any of the monoculture spheroids. Partially, this might be explained by the special arrangement of cells in the spheroid. Indeed, in the triculture spheroids the relative amount of the most well-transfected HaCaT and SK-MEL 28 cells was reduced in comparison to the monoculture spheroids. Furthermore, the CCD-1137Sk fibroblasts, which were already weakly transfected in the monoculture spheroids and present only in the outer cell layers, were enclosed in the triculture spheroids by several layers of HaCaT and SK-MEL 28 cells. From previous studies, it is also known that 3D cultures change cellular communication and thus also influence processes such as proliferation, protein biosynthesis, and mRNA expression [28,29]. Furthermore, while in the adherent cultures soluble factors released by any of the cells were likely available for all others due to the even exposure to media, the situation in the triculture spheroids might have been different. Indeed, in these cultures, fibroblasts are known to be largely separated from keratinocytes and melanoma cells, since they form a central core [14,15]. Thus, while cell–cell interactions would be expected to occur at the interface between the fibroblast core and the innermost layer of the keratinocyte ring, the interaction between fibroblasts and melanoma cells in this model are likely to be only weak via paracrine factors. Whether this contributed to the uptake behavior of the triculture spheroids, has remained elusive. In summary, the results indicate that substantial differences may be observed when using cell culture models of different complexity. Correlation with in vivo results will be necessary to allow for an assessment of the predictivity of the different model settings. The present data represent a step towards the development of such complex and indicative in vitro models. Such tools can be helpful for development of novel mRNA therapeutics, where direct delivery to the tumor is required.

## 5. Conclusions

In conclusion, this study allowed differences in the transfection efficiency of lipoplex nanoparticles on nonimmune cells in 2D and 3D melanoma models to be revealed. Transfection behavior of RNA nanoparticle formulations depended on several factors, such as the type of the cells, the cocultivation (interaction of different cells), as well as the 3D arrangement of the cells. In particular, while in 2D, transfection efficiency was similarly high for the different cell types—fibroblasts, keratinocytes, and melanoma cells—melanoma cells were better transfected in spheroids than fibroblasts. Furthermore, transfection was limited to outer cell layers in spheroids composed of either fibroblasts or keratinocytes, but it was rather uniform in melanoma spheroids. Finally, tricultures transfected well in 2D but poorly in spheroids, which was arguably due to the special arrangement of the different cell types in 3D. Therefore, in comparison to 2D models, the spheroid model as a platform to study 3D lipofection may provide a more realistic insight into the expected behavior in vivo.

## Figures and Tables

**Figure 1 cells-11-01026-f001:**
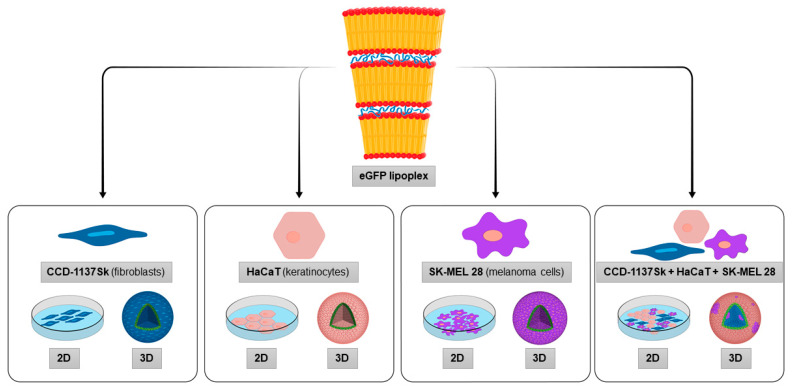
A systematic LPX transfection approach uses eGFP translation as a reporter for gauging effects of cell type, cellular interactions, and spatial constraints on transfection efficiency. Scheme of the experimental profile used in this study. Human CCD-1137Sk fibroblasts (blue), HaCaT keratinocytes (pink), and SK-MEL 28 melanoma cells (purple) were cultured in mono- or tricultures, and as adherent (2D) or spheroid cultures (3D) (lower part). Then, each cell culture was transfected with eGFP LPX (upper part and arrows). EGFP translation was evaluated by confocal fluorescence microscopy and quantitative analysis of eGFP cells and their distribution was performed (not shown).

**Figure 2 cells-11-01026-f002:**
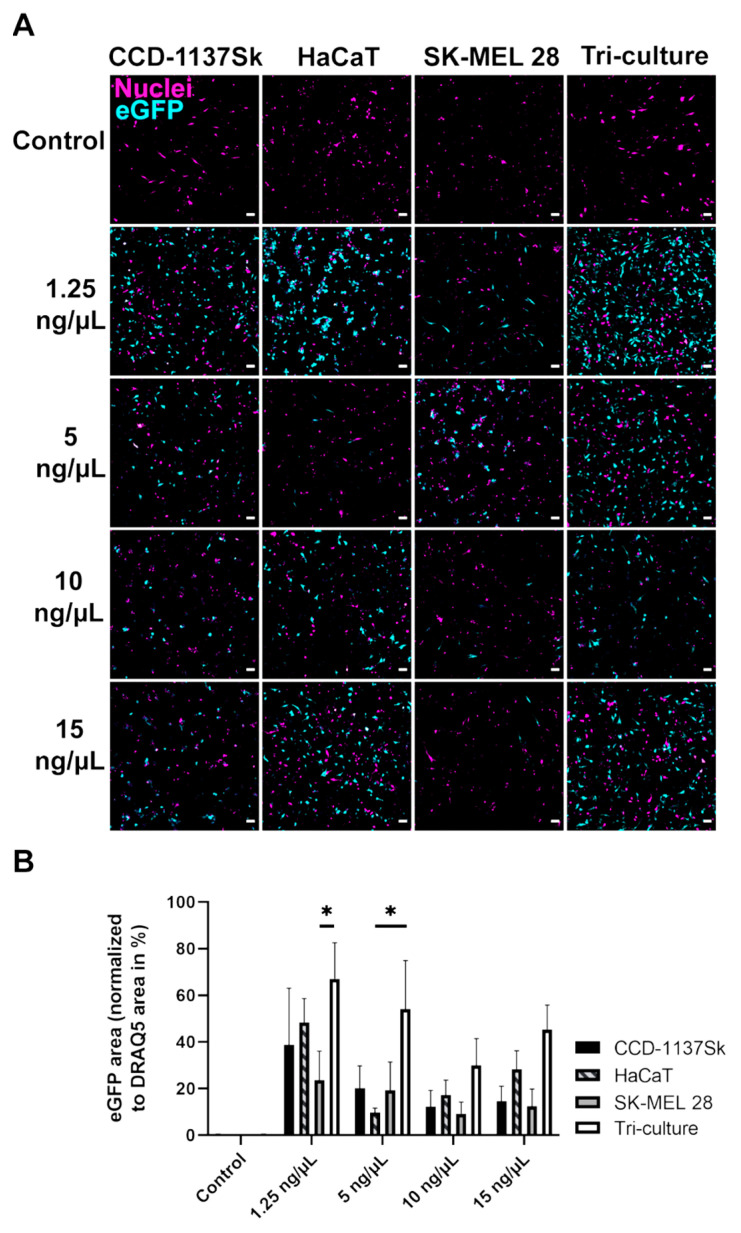
Lipoplex-mediated eGFP translation is enhanced in adherent tricultures of CCD-1137Sk, HaCaT, and SK-MEL 28 cells, compared to corresponding monocultures. For each culture condition, a total number of 3 × 10^4^ cells was seeded. The cell number and viability were determined by Vi-CELL XR. After 24 h, treatment with varying dosages of eGFP LPX or with PBS (control). Afterwards, cells were fixed and stained for nuclei. Confocal microscopy was used to visualize nuclei (magenta) and eGFP (cyan). (**A**) Representative confocal multi-tile scans are shown. Scale bars, 100 µm. (**B**) Quantitative analysis of the eGFP area normalized to nuclei signal area as a function of LPX amount added (0 to 15 ng/µL). Mean + SD (*n* = 3 independent experiments). Two-way ANOVA with Tukey’s multiple comparison post hoc test (α = 0.05). * *p* ≤ 0.05.

**Figure 3 cells-11-01026-f003:**
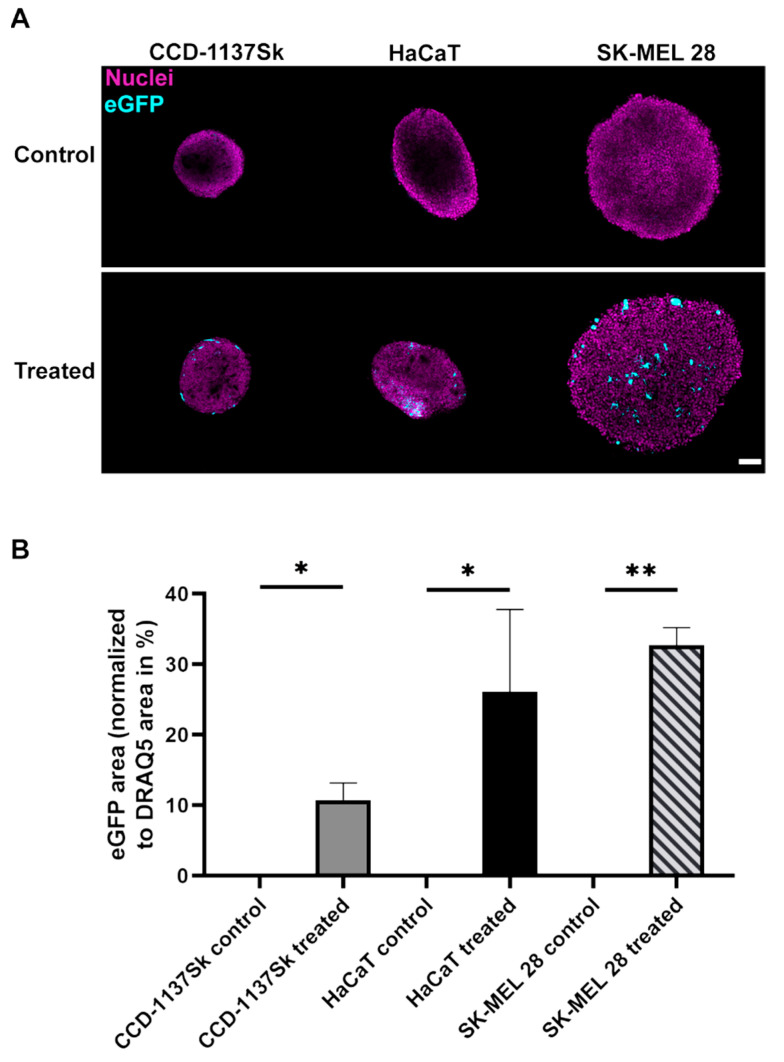
SK-MEL 28 spheroids show high expression of eGFP upon eGFP LPX transfection. For each spheroid, a total of 1 × 10^4^ cells was seeded and transfected with 10 ng/µL eGFP LPX. Control, PBS only. Confocal microscopy was used to visualize nuclei (magenta) and eGFP (cyan). (**A**) Representative single confocal sections in the center of representative spheroids are shown. Scale bar, 100 µm. (**B**) Quantitative analysis of the eGFP area normalized to nuclei signal area is depicted. Mean + SD (*n* = 3 independent experiments, at least 3 spheroids per condition). One-way ANOVA with Tukey’s multiple comparison post hoc test (α = 0.05). * *p* ≤ 0.05, ** *p* ≤ 0.01.

**Figure 4 cells-11-01026-f004:**
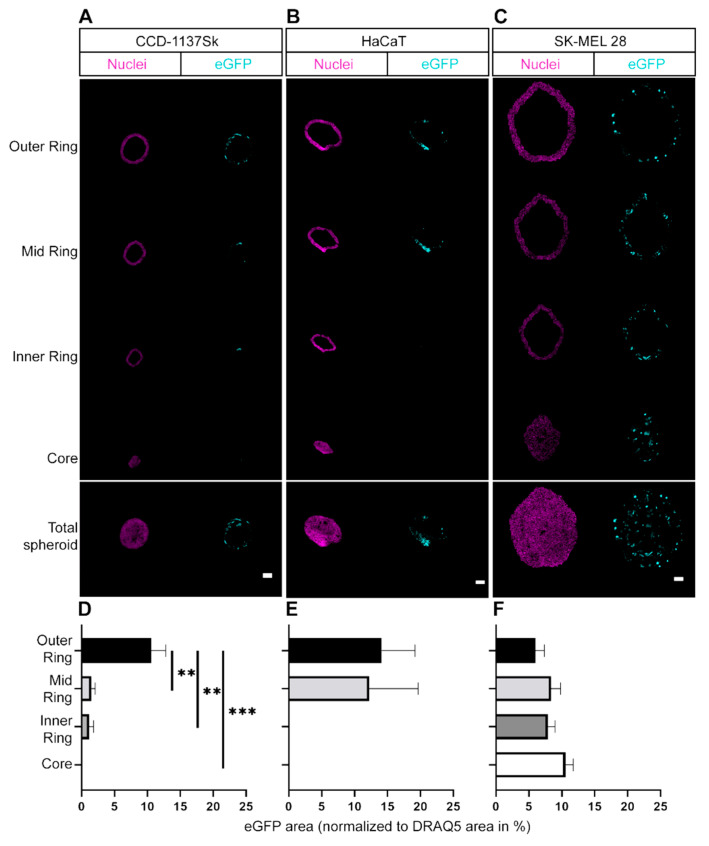
Differential distribution of eGFP-signals is observed in LPX-transfected 3D monoculture spheroids. For each spheroid, a total of 1 × 10^4^ cells were seeded and transfected with 10 ng/µL eGFP LPX. Spheroids were fixed and stained for nuclei. Confocal microscopy was used to visualize nuclei (magenta) and eGFP (cyan). (**A**–**C**) Representative confocal images of the spheroid segments (i.e., outer ring, mid ring, inner ring, and core) of single optical sections taken in the spheroid center (Total spheroid). Scale bars, 100 µm. (**D**–**F**) Quantitative analyses of eGFP area normalized to nuclei area for each segment of CCD-1137Sk (**D**), HaCaT (**E**), and SK-MEL 28 cells (**F**). Mean + SD (*n* = 3 independent experiments, at least 3 spheroids per condition). One-way ANOVA with Tukey’s multiple comparison post hoc test (α = 0.05). ** *p* ≤ 0.01, *** *p* ≤ 0.001.

**Figure 5 cells-11-01026-f005:**
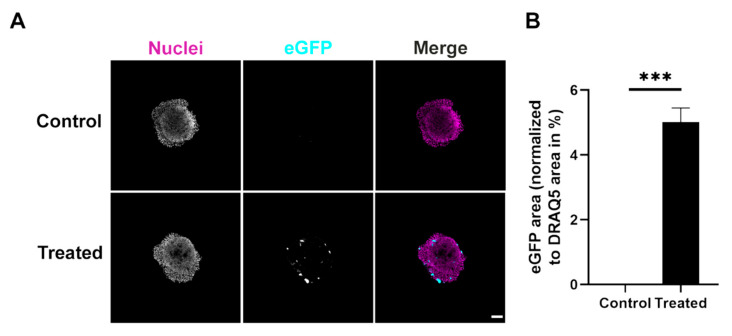
Triculture spheroids show reduced expression of the eGFP reporter protein compared to monocultures upon eGFP LPX transfection. For each triculture spheroid, identical amounts of cells were used and transfected with 10 ng/µL eGFP LPX (treated). Control, PBS only. Afterwards, spheroids were fixed and stained for nuclei (**A**). Shown are single confocal sections through the center of representative spheroids. In merge images, nuclei and eGFP are shown in magenta and cyan, respectively. Scale bar, 100 µm. (**B**) Quantitative analysis of the eGFP area normalized to nuclei area. Mean + SD (*n* = 3 independent experiments, at least 3 spheroids per condition). T test (two-tailed, unpaired, (α = 0.05), *** *p* ≤ 0.001).

**Figure 6 cells-11-01026-f006:**
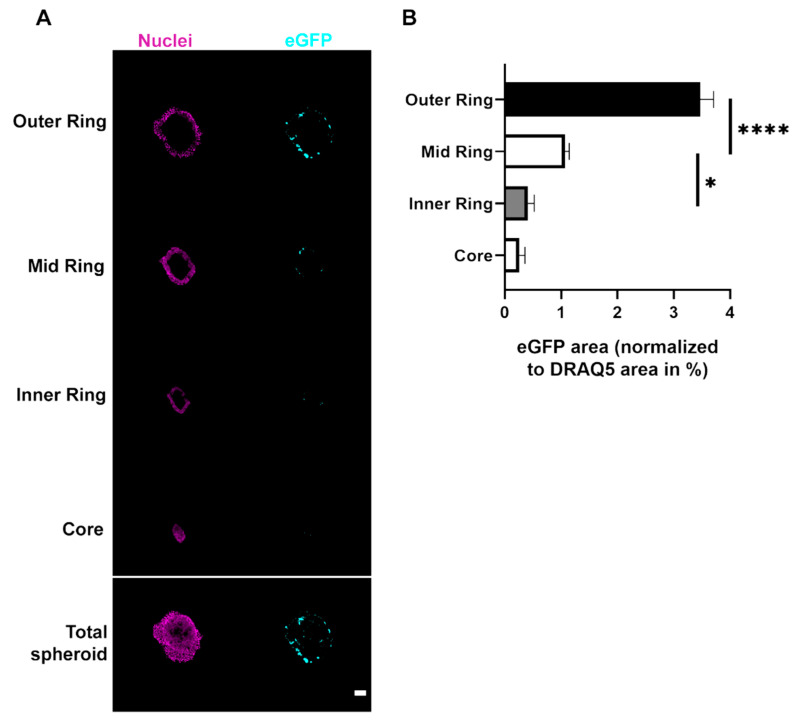
Triculture spheroids show highest expression of eGFP signal in the peripheral rings upon LPX transfection. For each triculture spheroid, identical amounts of cells were used and transfected with 10 ng/µL eGFP LPX, followed by fixation and staining of the nuclei. Confocal microscopy was used to visualize nuclei and eGFP at the maximum diameter of each spheroid. The segmentation and analysis were performed as in Figure 4. (**A**) Representative confocal images. Shown are whole optical sections (Total spheroid) and corresponding segments, i.e., outer ring, mid ring, inner ring, and core. Nuclei and eGFP, magenta and cyan, respectively. Scale bar, 100 µm. (**B**) Quantitative analysis. Graph shows the area of eGFP fluorescence normalized to nuclei area for each segment of triculture spheroids. Mean + SD (*n* = 3 independent experiments, 3 spheroids). One-way ANOVA with Tukey’s multiple comparison post hoc test (α = 0.05). * *p* ≤ 0.05, **** *p* ≤ 0.0001.

## Data Availability

All experimental data will be available upon request.

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
