# Peer review of "3D Melanoma Cocultures as Improved Models for Nanoparticle-Mediated Delivery of RNA to Tumors"

_cells, 2022, doi:10.3390/cells11061026_

Round 1

Reviewer 1 Report

The manuscript “3D Melanoma Co-Cultures as Improved Models for Nanoparti-2 cle-Mediated Delivery of RNA to Tumors” by Schäfer et al. presents a side-by-side comparison between 2D and 3D models of melanoma co-culture. The proposed 3D spheroids are presented as an in vitro tool with superior predictive capacity of the in vivo behavior of nanomedicines, compared to 2D traditional in vitro assays.

The introduction is very “essential” … the description of the disease (melanoma) minimal. In order to understand the "in vitro in vivo correlation", at least the main aspects of the in vitro model and of the clinical settings of the disease should be discussed.

There is also no information about the nanoparticles. For such a study, the physicochemical properties of the lipoplexes are almost as important as the 3D spheroids to understand their behavior in terms of internalization and penetration in the spheroids. Indeed, the 3D model was already characterized in previous publications (cited), and evaluated usig conventional cytotoxic agents. In this case, the novelty is in the therapeutic approach itself. The lack of information on the mRNA nanoparticles makes it poorly informative. Is the mRNA therapy at all a possibility for early stage melanoma therapy? besides immunotherapy, what are the potential targets of mRNA-lipoplexes?

Also, some of the figures are discussed in relation to other figures already published, but they are not correctly cited in the text, and this makes it difficult to follow.

Author Response

The manuscript “3D Melanoma Co-Cultures as Improved Models for Nanoparti-2 cle-Mediated Delivery of RNA to Tumors” by Schäfer et al. presents a side-by-side comparison between 2D and 3D models of melanoma co-culture. The proposed 3D spheroids are presented as an in vitro tool with superior predictive capacity of the in vivo behavior of nanomedicines, compared to 2D traditional in vitro assays.

The introduction is very “essential” … the description of the disease (melanoma) minimal. In order to understand the "in vitro in vivo correlation", at least the main aspects of the in vitro model and of the clinical settings of the disease should be discussed.

>>Response: We agree, the requested content was added at lines 70-78 and 81f.

There is also no information about the nanoparticles. For such a study, the physicochemical properties of the lipoplexes are almost as important as the 3D spheroids to understand their behavior in terms of internalization and penetration in the spheroids. Indeed, the 3D model was already characterized in previous publications (cited), and evaluated usig conventional cytotoxic agents. In this case, the novelty is in the therapeutic approach itself. The lack of information on the mRNA nanoparticles makes it poorly informative. Is the mRNA therapy at all a possibility for early stage melanoma therapy? besides immunotherapy, what are the potential targets of mRNA-lipoplexes?

>>Response: These points have now been addressed in lines 49-51 and lines 88-116.

Also, some of the figures are discussed in relation to other figures already published, but they are not correctly cited in the text, and this makes it difficult to follow.

>>Response: Most likely, this comment refers to the special distribution of fibroblasts, keratinocytes, and melanoma cells in tri-culture spheroids, that was briefly mentioned in lines 287-290 and 362-365. To better clarify the spatial distribution of the three cell types in tri-cultures, we now added a new figure S2 and a citation to the original reports on line 363.

Reviewer 2 Report

The paper is interesting and well written but additional experiments and details on methods are required.

Authors need to describe LPX preparation and which mRNA was used (commercial or not, modified or unmodified, encapuslation efficiency).

Authors should indicate the quantity of mRNA (µg) used to treat 2D and 3D cell cultures.

GFP expression should be quantitated on dissociated spheroids by flow cytometry.

Authors need to label the 3 cell types or use stable cell lines expressing fluorophores to establish the contribution/location of each cell type.

Authors should perform Ki67 staining on all spheroids types to visualize the location of proliferating cells.

Authors should correct "reference not found" through text.

Author Response

The paper is interesting and well written but additional experiments and details on methods are required.

Authors need to describe LPX preparation and which mRNA was used (commercial or not, modified or unmodified, encapuslation efficiency).

>>Response: These points have now been addressed in lines 88-116.

Authors should indicate the quantity of mRNA (µg) used to treat 2D and 3D cell cultures.

>>Response: These points have now been addressed in lines 88-116.

GFP expression should be quantitated on dissociated spheroids by flow cytometry.

>>Response: As requested, we carried out the analysis of reporter expression using flow cytometry. The novel data can be found in figure S3 and were referred to in lines 343f. Similar as in the microscopic analysis (Figure 3), Figure S3 shows that SK-MEL 28 cells had the highest levels of eGFP fluorescence upon LPX treatment. The other three spheroid types displayed only a minor increase in fluorescence. For fibroblasts and tri-cultures, this was in accordance with the microscopic data. However, for HaCaTs, a higher increase might have been expected. We think, that a likely explanation for this discrepancy between microscopic and cytometric data lies in the combination of the specific distribution of transfected cells and the preparation procedure for cytometry: indeed, it is not unlikely that the spheroid dissociation treatment led to a partial loss/cell death, in particular, of the outer spheroid cell layers. Further, while SK-MEL 28 cells were almost equally well transfected throughout the spheroid volume (see Figure 4), spheroids made of fibroblasts or keratinocytes as well as tri-culture spheroids showed eGFP-positive cells almost exclusively in the spheroid rim (see Figures 4 and 6). Thus, if these cells were particularly vulnerable during the dissociation procedure, the cytometry data might have delivered a skewed result in that respect. For now, we have left out this explanation in the current version of the revised manuscript, but it can be added if requested.

Authors need to label the 3 cell types or use stable cell lines expressing fluorophores to establish the contribution/location of each cell type.

>>Response: This aspect has now been added in figure S2 and referred to 290f. and 363f.

Authors should perform Ki67 staining on all spheroids types to visualize the location of proliferating cells.

>>Response: Ki67 staining has now been performed with all spheroid types. The novel data were included in figure S4 and referred to in lines 358f.

Authors should correct "reference not found" through text.

>>Response: Done.

Reviewer 3 Report

The results presented in the manuscript are interesting and can be recommended for publication. At the same time, it is necessary to dwell in more detail on the question of obtaining the complexes of mRNA and lipids themselves. In addition, the conclusions are written very schematically, please expand somewhat.

Author Response

The results presented in the manuscript are interesting and can be recommended for publication. At the same time, it is necessary to dwell in more detail on the question of obtaining the complexes of mRNA and lipids themselves. In addition, the conclusions are written very schematically, please expand somewhat.

>>Response: The first aspect was addressed in lines 88-116. The conclusions were expanded, see lines 393-399.

Round 2

Reviewer 2 Report

Authors adressed all comments.Pictures of Ki67 staining are very good.

I recommend accepting the revised paper.